# The Left Posterior Parietal Cortex Contributes to the Selection Process for the Initial Swing Leg in Gait Initiation

**DOI:** 10.3390/brainsci10050317

**Published:** 2020-05-22

**Authors:** Koichi Hiraoka, Shintaro Gonno, Ryota Inomoto

**Affiliations:** College of Health and Human Sciences, Osaka Prefecture University, Habikino City 583-8555, Japan; sbd02006@edu.osakafu-u.ac.jp (S.G.); sbd02004@edu.osakafu-u.ac.jp (R.I.)

**Keywords:** gait initiation, posterior parietal cortex, initial swing leg, decision making, selection

## Abstract

The present study examined whether the left posterior parietal cortex contributes to the selection process for the initial swing leg in gait initiation. Healthy humans initiated the gait in response to an auditory start cue. Transcranial magnetic stimulation (TMS) was given over P3, P4, F3 or F4 simultaneously, with the auditory start cue, in the on-TMS condition. A coil was placed over one of the four TMS sites, but TMS was not given in the off-TMS condition. The probability of right leg selection in the on-TMS condition was significantly lower than in the off-TMS condition when the coil was placed over P3, indicating that the left posterior parietal cortex contributes to the selection process of the initial swing leg of gait initiation. The latency of the anticipatory postural adjustment for gait initiation with the left leg was shortened by TMS over F4 or P4, but with the right leg was shortened by TMS over P3 or P4. Thus, the cortical process affecting the time taken to execute the motor process of gait initiation with the right leg may be related to the selection process of the initial swing leg of gait initiation.

## 1. Introduction

Healthy humans tend to consistently initiate gait with the same leg [1,2,3]. In contrast, in patients with Parkinson’s disease (PD) with the symptom freezing of gait (FOG), there is an inter-trial variability of the initial swing leg of gait initiation [1]. These contrasting findings indicate that impairment of the supraspinal central process causes an abnormal selection process of the initial swing leg of gait initiation. In other words, some specific cortical processes contribute to the selection process of the initial swing leg of gait initiation. Indeed, it was recently found that the supplementary motor area and cerebellum contribute to the anticipatory postural adjustment (APA) and ankle muscle activity of gait initiation [4]. In spite of this previous finding, the cortical sites contributing to the selection process of the initial swing leg of gait initiation have not yet been reported.

It has been shown in an animal experiment that the posterior parietal cortex (PPC) contributes to the selection of the effector [5]. The activity of the PPC is greater in the hemisphere contralateral to the hand that is moved [6]. More importantly, transcranial magnetic stimulation (TMS) over the left PPC increases the probability of left-hand use for unilateral reaching to a target [7]. This previous finding indicates that the left PPC contributes to the selection process of the hand for reaching.

Based on this previous finding, one may speculate that the left PPC also contributes to the selection process of the initial swing leg of gait initiation. However, this cannot be assumed to be true without validation because the mechanism underlying the selection process of the hand for reaching and the selection process of the initial swing leg of gait initiation are unlikely to be the same. That is, the PPC is specifically related to the motor process of visually guided reaching movement [8], but gait initiation is not a voluntary movement task; it is a postural task. In fact, the contingent negative variation is different between gait initiation and voluntary dorsiflexion of the ankle [9]. Based on this view, the research question of whether the left PPC contributes to the selection process of the initial swing leg of gait initiation or not is still a matter of investigation.

In the present study, an investigation was undertaken to determine whether the left PPC contributes to the selection process of the initial swing leg of gait initiation. This investigation involved giving TMS over P3, which is considered to be the site corresponding to the left PPC [10,11,12]. The control TMS sites were P4, F3, and F4. F3 and F4 are considered to be the sites over the dorsolateral prefrontal cortex [13] or over the middle frontal gyrus [14]. If the left PPC contributes to the selection process of the initial swing leg, the selection probability of the initial swing side can be expected to be changed by TMS over P3.

To investigate this issue, one methodological concern needed to be resolved. In healthy humans, the initial swing leg of gait initiation is consistent [1,2,3]. Thus, the opportunity to increase the use of the preferred initial swing leg by some intervention is much lower than that of the non-preferred initial swing leg. Based on this, one methodological concern relating to the effect of TMS on the selection of the initial swing leg is that the opportunity to change the selection probability of the initial swing side induced by TMS is unequal between the left and right. In order to address this methodological concern, equalizing the probability of left leg selection and the right leg selection is crucial.

The stance width of the start position changes the APA of gait initiation [15]. That is, the gait initiation is changed by positioning one foot forward or backward in relation to the other foot in the start position [16]. These findings indicate that the start position of the feet is an important parameter of gait initiation. The start position of the hands also changes the probability of hand selection for reaching [17], indicating that the hand position is an important parameter of the selection process of the hand for reaching. We extensively applied this view to our investigation of the selection process of the initial swing leg of gait initiation. That is, we supposed that changing the start position of the feet is a way to adjust the selection probability of the initial swing leg of gait initiation. Thus, in the present study, the non-preferred initial swing leg was placed behind the preferred initial swing leg in the start position, so that the probability of selecting the left leg and that of selecting the right leg for gait initiation was roughly equal. Taken together, the present study tested our hypothesis that the PPC contributes to the selection process of the initial swing leg during gait initiation.

## 2. Materials and Methods

### 2.1. Participants

The participants were 21 healthy humans aged 20.4 ± 0.1 years (15 males and 6 females). There was no neurological or orthopedic history in these participants. According to the Waterloo Footedness Questionnaire (WFQ) score [18,19], the preferred leg was right in 19 participants, was left in one participant, and was ambidextrous in another participant (Figure 1A). Informed consent was obtained from all participants. The experiment was approved by the Graduate School of Comprehensive Rehabilitation, Osaka Prefecture University Committee on Research Ethics (Approval Number; 2019-107).

### 2.2. Apparatus

A gravicorder measuring the center of pressure (COP) (1G06/I-B, Nihon Denshi Sanei, Tachikawa city, Tokyo, Japan) was placed on the floor. A walkway with a length of 180 cm and a width of 60 cm was placed in front of the gravicorder. TMS was delivered using a figure-of-eight coil (YM-133B; Nihon Kohden, Shinjuku district, Tokyo, Japan) connected to a magnetic stimulator (SMN-1200; Nihon Kohden, Shinjuku district, Tokyo, Japan). The maximum intensity of the coil was 0.96 T. There is a trade-off between focality and TMS intensity [20]. Thus, the intensity of the TMS was as low as 40% of maximum stimulator output to improve the focality of the stimulation. The sound pressure level of the coil click near the coil was about 48 decibels. An auditory start cue at a frequency of 1 kHz with 100 trains was given through earphones in the ears. The sound pressure level of the start cue in each earphone was 67 decibels. The coil was positioned away from the ears, but the earphones were placed in the ears. Thus, the sound pressure of the earphones was much greater than the coil click in the ears.

### 2.3. Default Gait Initiation Session

The preferred initial swing leg of gait initiation was determined in this session. The participants stood with their big toes 6 cm apart on the medial-lateral axis. Both feet were in the same position in terms of the anterior-posterior axis. They initiated the gait from this position in response to an auditory start cue and walked through to the end of the walkway (default gait initiation). When they reached the end of the walkway, they came back to the start position and maintained the quiet stance until the next start cue was delivered. The participants were asked to initiate the gait without paying attention to the initial swing side. The participants were not informed that an experimenter observed the initial swing leg of gait initiation.

Before conducting experimental trials, default gait initiation was performed once, and the baseline COP, which was the mean COP in the time window 100–0 ms before the start cue, was estimated. This baseline COP was considered to be the target COP. In the experimental trials, an experimenter monitored the online lissajous curve of the COP, where the medial-lateral COP (COPx) was on the horizontal axis and the anterior-posterior COP (COPy) was on the vertical axis, on a visual display, and triggered the start cue when the baseline COP was within ±1 cm of the target COP. When the baseline COP was out of the target range, the trial was discarded. The trials with an APA latency of longer than 300 ms were considered to be trials in which the participants did not pay attention to the start cue and were discarded. A warning cue was not delivered to rule out time prediction and to avoid the COP moving out of the target range in the interval between the warning and start cues.

The experimental trial was repeated 10 times. The discarded trials were retried after the end of those 10 trials. An experimenter visually identified the initial swing leg of gait initiation, and the number of trials in which they initiated the gait with the right leg was counted. The initial swing leg that was selected in more than 5 out of 10 trials was considered to be the preferred initial swing leg of default gait initiation.

### 2.4. Tested Feet Position Session

The start feet position for the trials in the TMS session (tested feet position) was determined in this session. The participants initiated the gait in response to an auditory start cue. In the start position, they placed the foot of the non-preferred initial swing leg behind the other foot. The distance between the left and right big toes in the anterior-posterior axis was named the feet distance (Figure 1C). The value of the feet distance was positive when the left foot was behind the right foot and was negative when the right foot was behind the left foot. In each feet distance, gait initiation was performed once before the experiment, and the target COP was estimated. In the experimental trials, the experimenter monitored the lissajous curve of the COP and triggered the start cue when the baseline COP was within ±1 cm of the target COP. The experimental trial was repeated 10 times (trial block). The discarding criteria were same as in the default gait initiation session.

The feet distance was adjusted by 0.5–1 cm increments in each trial block to determine the feet distance where the probability of left leg selection and right leg selection was about equal (the range was within 0.4–0.6 of the probability of right leg selection). This process continued until an appropriate feet distance, in which the probability of gait initiation with the left leg and with the right leg was about equal, was identified. This feet position was named the tested feet position. The feet distance in the tested feet position was named the tested feet distance. The participants were not informed that an experimenter observed the initial swing leg of gait initiation.

### 2.5. TMS Session

The effect of the TMS on the probability of right leg selection for the initial swing leg of gait initiation was determined in this session. The participants who preferentially initiated the gait with the left leg in the default gait initiation session, and the participants for whom the tested feet position was impossible to determine in the tested feet position session, were excluded in this session. The reason for excluding the participants who preferentially initiated the gait with the left leg was to unify the response side and minimize the response variability, because of the response asymmetry during gait initiation [2]. Before beginning this session, an experimenter confirmed that the participants did not perceive the coil click explicitly when the auditory start cue was simultaneously delivered, so that the coil click did not affect the process of gait initiation. The tested feet position, determined in the tested feet position session, was used as the start position of gait initiation. Gait initiation from the tested feet position was named the tested gait initiation.

A coil was placed over F3, P3, F4 or P4. P3 is considered to be the site over the left PPC, and P4 is considered to be the site over the right PPC [10,11,12]. The participants wore a swim cap. The edge of the swim cap was firmly fixed by elastic bandages over the forehead, dorsal neck, and ears. The TMS sites were determined by measuring the distance from the inion, nation, and ears, in accordance with the international 10–20 system in each participant, and the sites were marked over the swim cap to enable the experimenter to hold the coil at the stimulus site consistently and accurately. The orientation of the coil was along the rostro-caudal direction, to induce a posterior-anterior electrical current in the brain [7].

Before conducting the trials, an experimenter asked the participants to ignore the coil and TMS, and not to pay attention to the initial swing side of gait initiation. The participants were not informed that an experimenter observed the initial swing leg of gait initiation. Before conducting the experimental trials, the tested gait initiation was performed once, and the target COP was estimated from this trial. In the experimental trials, the experimenter monitored the lissajous curve of the COP and triggered an auditory start cue when the COP was within ±1 cm of the target COP. Another experimenter stood behind the participant quietly. The coil was held by this experimenter and was placed over the target site of the scalp very softly immediately before the start cue, so that the coil did not interfere with gait initiation. The participants initiated the gait in response to the start cue and walked through to the end of the walkway. When they reached the end of the walkway, they came back to the start position and maintained the quiet stance until next start cue was delivered.

In the on-TMS condition, TMS was given simultaneously with the start cue. It has been reported that preparatory activity is processed before the start cue, because startle auditory stimulation influences the APA latency before the start cue [21]. In contrast, corticospinal excitability in the tibialis anterior muscle is gradually increased 100 ms after the start cue or later. Based on this previous finding, TMS was delivered simultaneously with the start cue so that the TMS influenced the preparatory activity, including response selection, before beginning motor execution, during which corticospinal excitability gradually increased.

In the off-TMS condition, the coil was placed over one of the four sites used in the on-TMS condition, but TMS was not given. In each trial, either the on-TMS or off-TMS condition was assigned with the coil position at either F3, P3, F4 or P4. When the quiet stance is maintained by participants consistent and accurate TMS is possible when the coil is manually held by an experimenter, according to previous studies [22,23]. When the start cue and TMS were given simultaneously, the participants still maintained the quiet stance in the present study. Accordingly, TMS was consistently and accurately given to the target site. Neither the experimenter holding the coil nor the participants were informed of whether TMS was given or not in the forthcoming trial.

Each of the eight conditions (4 coil positions × 2 TMS conditions) consisted of 20 trials. A total of 160 successful trials were obtained in the TMS session. The coil position and on/off state of the TMS were randomly altered trial by trial. Thus, the presence or absence of TMS in the forthcoming trial was not predictable for the participants, or for the experimenter who held the coil. In a preliminary experiment, we confirmed that the probability of the initial swing leg selection changed as the number of the trials increased. Thus, in the TMS session, we monitored and adjusted the probability of the leg selection through the following procedure. One hundred and sixty trials were assigned into four experimental blocks, and each block consisted of 40 trials. Five trials in each condition were conducted in a block. We confirmed the equal selection probability of the initial swing leg between the left and right in the most recent 10 trials in each block. That is, in each block, trials were conducted until the probability of right leg swing was within 0.4–0.6 in the most recent 10 trials. Those most recent 10 trials, in which the probability of the right leg swing was within 0.4–0.6, were considered to be the first 10 trials of the block, and 30 trials without monitoring the probability were additionally conducted after those 10 trials. This procedure ensured the equal probability of leg selection in each block. The discarding criteria of the trials were the same as the tested feet position session. The discarded trials were retried after the end of each block.

### 2.6. Data Analysis

The probability of right leg selection across 20 trials in each condition in the TMS session was estimated. The baseline COP was subtracted from the target COP to estimate the deviation of the baseline COP from the target COP. The COP displacement in the S1 phase (first displacement of the COP to the backward and initial swing side) is considered to be the APA of gait initiation [24]. Based on this, the latency of the displacement onset of the COPy (the onset of the S1 phase) was estimated (APA latency).

A two-way repeated measures ANOVA was conducted to test the main effect of the TMS site (P3, P4, F3, and F4) and that of the TMS. If the two-way ANOVA revealed a significant interaction between the main effects, then a test of the simple main effect was conducted on each level of another main effect. If our hypothesis, that PPC contributes to the selection process of the initial swing leg, is true, the two main effects would interact and the simple main effect of the TMS would be present only over P3. A one-way repeated measures ANOVA was conducted to test the difference in the mean APA latency between the tested gait initiation in the off-TMS condition and the default gait initiation. If the one-way ANOVA revealed a significant difference, then the multiple comparison test (Bonferroni’s test) was conducted. The one-way ANOVA was to test whether the time taken to initiate the gait is increased in the tested gait initiation, during which competition between the motor plan of the gait initiation with the left leg and that with the right leg is present. If our hypothesis is true, the APA latency of the tested gait initiation, with equal selection probability between the left and right leg, would be longer than the default gait initiation in which the participants initiate the gait mostly with the preferred initial swing leg. The result of the Greenhouse–Geisser’s correction was reported whenever Mauchly’s test of sphericity was significant. A paired t-test was conducted to test the difference in the baseline COP deviation between gait initiation with the left leg and that with the right leg. One-sample t-tests were conducted to test whether the probability of the right leg selection in each TMS site in the off-TMS condition was different from 0.5, to confirm that the probability of right leg selection in the off-TMS condition was not significantly different from the theoretical equal probability of leg selection between the left and right. Spearman’s rank correlation coefficient was estimated to test the relationship between the two measures. The alpha level was 0.05. The data were expressed as the mean and standard error of the mean.

## 3. Results

### 3.1. Preferred Initial Swing Leg

The preferred initial swing leg of the default gait initiation was left in three participants, was ambidextrous in one participant, and was right in the other 17 participants (Figure 1B, Table 1). In 13 out of 21 participants, they initiated the gait with the same leg across 10 trials. There was no significant correlation between the probability of right leg selection of the default gait initiation and the WFQ score (ρ = −0.07, *p* = 0.770). This indicates that the leg preference and the preferred initial swing leg do not share the same process.

### 3.2. TMS Effect on Leg Selection

In six participants, the tested feet position was not found in the tested feet position session. In those participants, one initiated the gait with the left leg in 10 out of 10 trials at any feet distance, two initiated the gait with the left leg in 10 out of 10 trials at any feet distance, and three initiated the gait with the left leg in 10 out of 10 trials at some feet distances, but initiated the gait with the right leg in 10 out of 10 trials at other feet distances. These participants were excluded from the TMS session. In addition, two participants who preferentially selected the left leg during default gait initiation were excluded from the TMS session. This means that the TMS session was conducted with a total of 13 participants (Table 1). In 2 out of the 13 participants, the tested feet position was slightly changed (0.5 cm) in one of the four trial blocks to maintain the equal probability of the initial swing leg being the left or right leg. None of the participants perceived the coil click induced by TMS explicitly when the auditory start cue was simultaneously delivered. This means that they initiated the gait without perceiving the coil click.

The probability of right leg selection for each TMS site of the off-TMS condition was not significantly different from the theoretical equal probability of leg selection (0.5) (Figure 2). A two-way ANOVA on the probability of right leg selection did not reveal a significant main effect of TMS (F(1, 12) = 1.333, *p* = 0.271, η^2^ = 0.00) and TMS site (F(1.31, 15.76) = 0.350, *p* = 0.621, η^2^ = 0.02) (Greenhouse-Geisser correction; Mauchly’s Test, *p* < 0.001). There was a significant interaction between the main effects (F(3, 36) = 3.024, *p* = 0.042, η^2^ = 0.02). The test of the simple main effect revealed that the probability of right leg selection in the on-TMS condition was significantly smaller than that in the off-TMS condition when the coil was placed over P3 (F(1, 48) = 9.466, *p* = 0.004). In contrast, such a significant simple main effect was not observed when the coil was positioned over F3 (F(1, 48) = 0.278, *p* = 0.600), F4 (F(1, 48) = 0.193, *p* = 0.662), or P4 (F(1, 48) = 0.626, *p* = 0.433). The test of the simple main effect did not reveal a significant main effect of the TMS site in either the on-TMS (F(3, 72) = 0.549, *p* = 0.650) or off-TMS condition (F(3, 72) = 0.717, *p* = 0.545). The effect of TMS over P3 on the probability of right leg selection, expressed as the probability in the on-TMS condition subtracted by the probability in the off-TMS condition, was not significantly correlated with the WFQ score (ρ = −0.20, *p* = 0.512) or with the probability of right leg selection for default gait initiation (ρ = 0.23, *p* = 0.457). These findings indicate that the effect of TMS over P3 is not related to leg preference or the preferred initial swing leg. The effect of TMS over P3 on the probability of right leg selection was not significantly correlated with the tested feet distance (ρ = 0.37, *p* = 0.218). This finding did not support a view that the extent between the foot of the non-preferred initial swing leg and that of the preferred initial swing leg in the anterior-posterior direction affects the effect of the TMS over the P3 on the probability of right leg selection.

### 3.3. APA Latency

The effect of TMS and the TMS site on the APA latency was tested in the trials of gait initiation with the left leg and in those with the right leg in the TMS session. For the participants included in the TMS session, two participants were excluded from the analysis because trials with the left leg swing or trials with the right leg swing were absent in some conditions. Thus, the APA latency was analyzed for 11 participants (Table 1).

In the trials in which the participants initiated the gait with the left leg, there was no significant main effect of TMS (F(1, 10) = 4.818, *p* = 0.053, η^2^ = 0.02) and the TMS site (F(1.90, 19.00) = 0.101, *p* = 0.896, η^2^ = 0.00) (Greenhouse–Geisser correction; Mauchly’s Test, *p* = 0.039), with a significant interaction between the main effects (F(3, 30) = 3.015, *p* = 0.045, η^2^ = 0.02) (Figure 3A). The test of the simple main effect revealed that TMS over F4 (F(1, 31) = 9.132, *p* = 0.005) or P4 (F(1, 31) = 4.945, *p* = 0.034) significantly decreased the APA latency. In the trials in which the participants initiated the gait with the right leg, there was a significant main effect of TMS (F(1, 10) = 23.894, *p* < 0.001, η^2^ = 0.04), but there was no significant main effect of the TMS site (F(3, 30) = 2.619, *p* = 0.069, η^2^ = 0.02), with a significant interaction between the main effects (F(3, 30) = 4.792, *p* = 0.008, η^2^ = 0.03) (Figure 3B). The test of the simple main effect revealed that TMS over P3 (F(1, 40) = 20.707, *p* < 0.001) or P4 (F(1, 40) = 12.773, *p* < 0.001) significantly decreased the APA latency.

The difference in the APA latency between the tested gait initiation in the off-TMS condition and the default gait initiation was investigated (Figure 4). Thirteen participants were included in this analysis (Table 1). In the default gait initiation, the participants initiated the gait from an even feet position, but in the off-TMS conditions, they initiated the gait from the tested feet position, in which the probability of the initial swing leg was about equal between the left and right. Thus, this comparison allowed us to test whether competition between the motor plan of gait initiation with the left leg and that with the right leg increases the time taken to initiate the gait. A one-way ANOVA revealed a significant difference in the APA latency among the conditions (F(2.30, 27.60) = 5.956, *p* = 0.005, η^2^ = 0.01) (Greenhouse–Geisser correction; Mauchly’s Test, *p* = 0.024). A multiple comparison test revealed that the APA latency of the tested gait initiation (off-TMS condition) was significantly longer than that of the default gait initiation (*p* < 0.05).

### 3.4. Tested Feet Position

In 6 out of 21 participants, a tested feet position with an equal probability of the initial swing leg being the left or right leg was not found. The tested feet distance in the other 15 participants was 3.2 ± 1.4 cm (Figure 1C, Table 1). There was a significant positive correlation between the tested feet distance and the probability of right leg selection in default gait initiation in those 15 participants (ρ = 0.82, *p* < 0.001, Figure 1D). This indicates that the extent of the backward placement of the non-preferred initial swing leg causing the equal probability of the initial swing leg being the left or right leg is higher as the right leg preference is greater.

### 3.5. Deviation of Baseline COP

In order to rule out the possibility that a deviation of weight distribution between the feet is the cause of the change in the probability of leg selection, the deviation of the baseline COP from the target baseline COP was tested in the trials of gait initiation with the left leg and with the right leg. The mean deviation of the baseline COP from the target COP was within 1 mm (Figure 5). There was no significant difference in the deviation of the baseline COP between gait initiation with the left leg and that with the right leg in both axes (COPx; *p* = 0.091, COPy; *p* = 0.727, paired *t*-tests).

## 4. Discussion

In the present study, we hypothesized that TMS, particularly over P3, could change the selection of the initial swing leg for gait initiation, if the left PPC contributes to the selection process of the initial swing leg of gait initiation. In line with our hypothesis, TMS, particularly over P3, significantly reduced the probability of right leg selection for gait initiation.

### 4.1. Contribution of the Left PPC

The most likely explanation for the change in the probability of leg selection induced by TMS over P3 is that the change in the cortical activity under P3 biased the selection process of the initial swing leg of gait initiation. P3 is considered to be the site over the left PPC [10,11,12]. Thus, our hypothesis, that the left PPC contributes to the selection process of the initial swing leg of gait initiation, is the most likely explanation for the effect. The present finding was consistent with previous findings on the selection process of the hand for reaching; TMS over the left PPC caused a significant decrease in right hand use [7]. Thus, the left PPC is likely the common site contributing to the selection process of both the initial swing leg of gait initiation and the hand for reaching.

The left hemisphere-dominant contribution of the PPC to the motor process has previously been reported. For example, motor attention is dominantly processed in the left PPC [25]. A difference in the activation of the intraparietal sulcus between choice and simple reaction time tasks is present only in the left PPC [26], indicating that the selection process is predominantly processed in the left PPC. Accordingly, the particular contribution of the left PPC to the selection process of the leg for gait initiation may be explained by this asymmetrical role of the PPC.

### 4.2. Limb Selection and Motor Execution

A monaural auditory start cue to the ear ipsilateral to the non-preferred initial swing leg changed the probability of the initial swing leg when the cue side was not predictable [3]. The change in the selection process of the initial swing leg must have been processed within the reaction time in this previous study because the monaural auditory cue was given as the start cue. A motor execution process is implemented in the reaction time. Accordingly, the selection process of the initial swing leg is modulated during the execution process of gait initiation. In the present study, TMS was given simultaneously with the start cue. Thus, the change in the selection process of the initial swing leg induced by TMS over the left PPC occurs during the execution process. There are two possible mechanisms underlying this finding. One is that the selection of the limb can be changed even during the execution process. The other is that the selection and execution processes are not sequential but rather parallel processes, where both processes run simultaneously.

### 4.3. APA Latency

TMS reduced the reaction time of the motor response no matter which TMS site was used, and this site-non-specific TMS effect on the reaction time can be explained by intersensory facilitation [27]. In contrast, in the present study, the APA latency during gait initiation with the left leg was shortened by TMS either over P4 or F4, but with the right leg was shortened by TMS either over P3 or P4. That is, the TMS effect on the APA latency was site specific. Thus, the reduction in the APA latency in the present study is not explained by intersensory facilitation.

Oliveira and colleagues supposed that the left PPC contributes to the motor process of both hands for reaching, but the right PPC unilaterally contributes to the process of the left hand for reaching [7]. If this view is true, when the activity of the left PPC is interfered with by TMS, then compensation of the left PPC activity by the right PPC is not possible. In this situation, the decrease in the selection of the right hand for reaching induced by TMS over the left PPC is explained by the view that the right PPC cannot compensate for the interfered activity of the left PPC, because the right PPC contributes unilaterally to the left hand reach.

One may speculate that this view is applicable for the explanation of the change in the APA latency induced by TMS during gait initiation. When postural perturbation is induced, the APA latency increases with the delay of the stepping movement, indicating that the APA and stepping movement involve a common motor process [28]. Based on this view, the motor process of gait initiation, including the APA and step movement, occurs within the reaction time of the APA, and the APA latency reflects the time taken to execute the motor process of gait initiation. The direction of the COP displacement in the medial-lateral axis is contrary between gait initiation with the left leg and that with the right leg [2,3,29]. Thus, the APA latency of gait initiation with the left leg represents the time taken for execution of the motor process of gait initiation with the left leg, and vice versa.

In the present study, TMS over the left PPC unilaterally shortened the APA latency of the right leg swing, but TMS over the right PPC bilaterally shortened the APA latency of the left and right leg swing. This finding indicates that the right PPC, under P4, contributes to the time taken for the motor process of gait initiation either with the left or right leg, but the left PPC under P3 contributes to the time taken for the motor process of gait initiation solely with the right leg. That is, the right PPC is able to compensate for the interfered activity of the left PPC induced by TMS over P3. Accordingly, the decrease in right leg selection induced by TMS over the left PPC in the present study cannot be explained by the deficit in the compensatory activity of the right PPC for the interfered activity of the left PPC.

The most likely explanation for the specific effect of TMS over the left PPC on the initial swing leg selection is the unilateral contribution of the left PPC to the time taken for the motor process of gait initiation with the right leg. TMS over F4 decreased the APA latency of gait initiation with the left leg, and that over P4 decreased the latency of gait initiation with both the left and right legs, but that over P3 decreased the latency of gait initiation solely with the right leg. That is, P3 is the particular site where TMS changes solely the time taken for the motor process of gait initiation with the right leg. Thus, the particular effect of TMS over the left PPC on the selection process of the initial swing leg is explained by the cortical process affecting solely the time taken to execute the motor process of gait initiation with the right leg, which is related to the selection process of the initial swing leg of gait initiation.

One concern about the findings regarding the APA latency is that the change rate of the latency was as small as around 7%, although this change was significant. In a previous study, the APA latency of gait initiation was changed between the go and switch (change in the initial swing leg during the time between the start cue and APA onset) tasks, or between the go and stop (withholding gait initiation before the onset of the APA) tasks, and the change rate was 8–10% [29]. In this previous study, the change in the APA latency was due to a change in the process of the initial swing leg selection, motor execution, or inhibition. In the present study, TMS was given to change the motor and/or selection process of the initial swing leg. Thus, the change in the central motor status may change the APA latency by a small amount—around 7–10%.

### 4.4. Coil Position

In the present study, the coil was placed over four sites (F3, F4, P3 and P4). Thus, one may speculate that the coil position may have influenced the probability of right leg selection for gait initiation. That is, the initial swing leg may have been selected before the start cue, and this selection may have been influenced by the position of the coil placed before the start cue. The present findings did not support this speculation. There was a significant interaction between the two main effects (TMS and TMS site), but the test of the simple main effect did not reveal a significant effect of the coil position on the probability of right leg selection in the off-TMS condition.

### 4.5. Coil Click

One possible explanation for the effect of TMS over P3 on the selection process of the initial swing leg is the coil click produced by TMS. The reaction time of the hand response decreases when TMS is given either over the motor cortex, Cz, or Pz, or even if it is given in the air near the scalp [27]. This indicates that the coil click has a non-site-specific effect on the motor process. The coil click in the air near the scalp causes an auditory-evoked potential [30]. Accordingly, the non-site-specific effect of TMS is likely explained by the coil click inducing auditory-related cortical activity.

The coil click can affect the selection process of the limb, according to previous findings that asymmetrical auditory input changes the selection process of the limbs. For example, the probability of hand selection is affected by asymmetrical auditory input [31]. In addition, an asymmetrical auditory start cue has been found to influence the selection process of the initial swing leg of gait initiation [3]. Thus, one may speculate that the change in the probability of the initial swing leg of gait initiation induced by TMS over P3 could be explained by the asymmetrical auditory input caused by the coil click over the left scalp.

However, the coil click was not the likely cause of the effect of TMS over P3 on the selection process in our findings. Firstly, the sound pressure of the auditory start cue was so great that the participants did not perceive the coil click explicitly, meaning that the coil click was well masked by the auditory start cue. Secondly, the effect of TMS on the probability of right leg selection was not side dependent, but site specific. The coil click was produced near the left ear when TMS was given over F3 or P3, causing relatively greater auditory input to the left ear. In contrast, the coil click was produced near the right ear when TMS was given over F4 or P4, causing relatively greater auditory input to the right ear. Thus, if the coil click was the cause of the change in the selection of the initial swing leg, the effect of the coil click would be dependent on whether TMS was given over the left or right scalp. TMS over the left cortex (F3 and P3) and over the right cortex (F4 and P4) did not cause a symmetrical effect on the probability of right leg selection; the effect was specifically induced by TMS over P3.

### 4.6. Weight Distribution

The change in the weight distribution between the feet modulated the medial-lateral amplitude of the APA of gait initiation [32]. The duration of the APA before gait initiation was dependent on the weight distribution between the feet in the start position [33]. These previous findings indicate that weight distribution between the feet is the important determinant of the APA of gait initiation. Based on this view, one may speculate that the baseline COP may influence the selection process of the initial swing leg.

To rule out this concern, an experimenter monitored the baseline COP, and triggered the start cue when the baseline COP was around the target COP. Immediately after each trial, the baseline COP of the trial was estimated, and the trial was discarded if the baseline COP was not within the target range. As a result, the mean deviation of the baseline COP from the target COP was within 1 mm (see Section 3.5 and Figure 5). This indicates that the gait initiation was performed from the start position around the target COP. In addition, there was no significant difference in the deviation of the baseline COP from the target COP between the trials in which the gait was initiated with the left leg and the trials in which it was initiated with the right leg. This indicates that the selection probability of the initial swing leg between the left and right is not dependent on the baseline COP. Taken together, the effect of TMS over P3 on the selection process of the initial swing leg is not caused by a deviation of the weight distribution between the feet.

### 4.7. Unequal Probability of the Swing Side

In most participants, the probability of the initial swing leg of gait initiation was unequal between the left and right legs during default gait initiation. This inequality causes an unequal opportunity for change of the initial swing side. For example, if one always initiates the gait with the right leg, the opportunity to increase the probability of right leg selection is zero. To remove this inequality, the probability of left leg selection and that of the right leg selection were equalized by placing the foot of the non-preferred swing side behind the other foot. As a result, the selection probability of the initial swing leg in the off-TMS condition was not significantly different from the theoretical equal probability (0.5), indicating that the probability of the swing side of gait initiation was roughly equal between the left and right leg. Thus, the unequal opportunity for change in the initial swing leg is not the likely cause of the effect of TMS over P3 on the selection process of the initial swing leg.

### 4.8. TMS Effect and Preferred Initial Swing Leg

In a previous study, the correlation coefficient of the TMS effect over the left PPC on hand selection and hand preference was estimated based on the hypothesis that the effect of TMS over the left PPC on hand selection is due to the interference of the selection process by hand preference [7]. The result was negative; a significant correlation coefficient was not found. We also estimated the correlation coefficient based on a similar hypothesis that the change in the initial swing leg selection induced by TMS over P3 is due to the interference of the selection process by the initial swing leg preference. Consistent with the previous findings, the effect of TMS over P3 on the probability of right leg selection was not significantly correlated with the probability of right leg selection in default gait initiation. Thus, the change in the selection of the initial swing leg induced by TMS over P3 is not associated with the interference of the selection process by the initial swing leg preference.

### 4.9. Competition of Motor Plans

In previous studies, the latency of hand movement increased when the participants reached for a target, wherein the probability of left hand selection and right hand selection was roughly equal [7,31]. Similarly, in the present study, the APA latency of gait initiation from the tested feet position, where the probability of leg selection was equalized, was significantly longer than that of default gait initiation. When the probability of leg selection was about equal between the left and right limbs, both the motor process of the left limb and that of the right limb must have been prepared. The increase in the reaction time in this situation is explained by the competition between the motor process of the left limb and that of the right limb [7]. According to the bounded accumulation model, selecting one from the two effectors is achieved by the process of accumulating information over time [34]. Then, the decision process drifts to one of the two response boundaries with the accumulation of information, and the decision is made when the accumulation level surpasses the boundary. Based on this model, the increase in the APA latency in the tested gait initiation compared with that in the default gait initiation in the present study is explained by both motor processes being away from the boundary of the decision making, due to the equal competition between the two motor plans.

In PD patients with FOG, there is an inter-trial variability of the initial swing leg of gait initiation [1]. Based on the bounded accumulation model, this is also explained by the motor plans being prepared and competing with each other before gait initiation. The decision does not consistently drift to one direction trial by trial in PD patients with FOG. Accordingly, the tested gait initiation, during which the probability of the initial swing leg is equal, and the gait initiation of PD patients with FOG, may share the same selection process of the initial swing leg.

### 4.10. Excluded Participants

In 6 out of 21 participants, equal probability of the initial swing leg was not achieved by placing the non-preferred initial swing leg behind the other. Accordingly, susceptibility to changes in feet position was variable across the participants. Our speculation regarding this finding is that the selection process of the initial swing leg was insensitive to change in the feet position, or the range of the feet distance causing the equal probability of the initial swing side was exceptionally narrow, in those six participants. This is an interesting issue to be investigated in future studies.

### 4.11. Limitations

The coil was held by an experimenter manually, but evidence proving that the coil was positioned precisely over the TMS site was not obtained. This was a weakness of this study. The mean probability of the right leg selection in the off-TMS of P3 condition tended to be greater than that of the other sites, although this tendency was statistically insignificant. If this probability is truly greater than for the other sites, the present findings regarding the effect of TMS on P3 is possible to be considered due to the increase in the probability of the off-TMS of P3. In the present study, 20 participants were recruited, but only 13 participants were included in the data analysis of the TMS session. Thus, we cannot rule out the possibility that this relatively small sample size, which weakens the statistical power, resulted in an insignificant effect being found for the TMS site in the off-TMS condition.

## 5. Conclusions

TMS over P3 increased left leg selection for the initial swing leg of gait initiation. This indicates that the left PPC contributes to the selection process of the initial swing leg of gait initiation. TMS over P3 was the particular conditioning TMS that decreased the latency of gait initiation with the right leg. Accordingly, the selection process of the initial swing leg of gait initiation may be mediated by the cortical process affecting the time taken to complete the motor execution process of gait initiation with the right leg.

## Figures and Tables

**Figure 1 brainsci-10-00317-f001:**
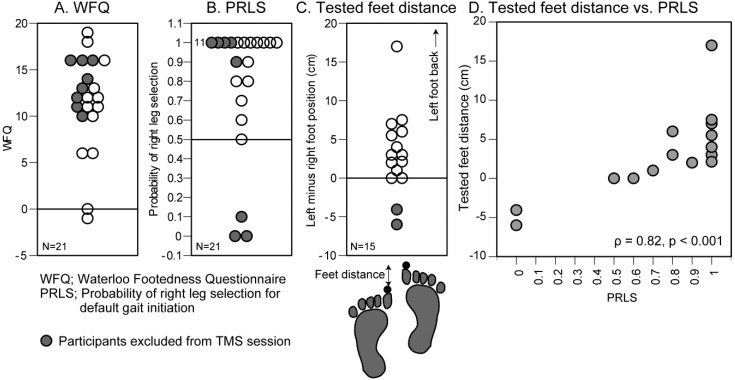
Waterloo Footedness Questionnaire (WFQ) (**A**), probability of right leg selection (PRLS) of the default gait initiation (**B**), the tested feet distance (**C**), and the plots of the tested feet distance as relation with the PRLS (**D**). Each data point indicates one participant. Filled circles indicate the participants excluded from the TMS session. The number in panel B indicates the number of participants who initiated the gait with the right leg across all trials (**B**). TMS: transcranial magnetic stimulation.

**Figure 2 brainsci-10-00317-f002:**
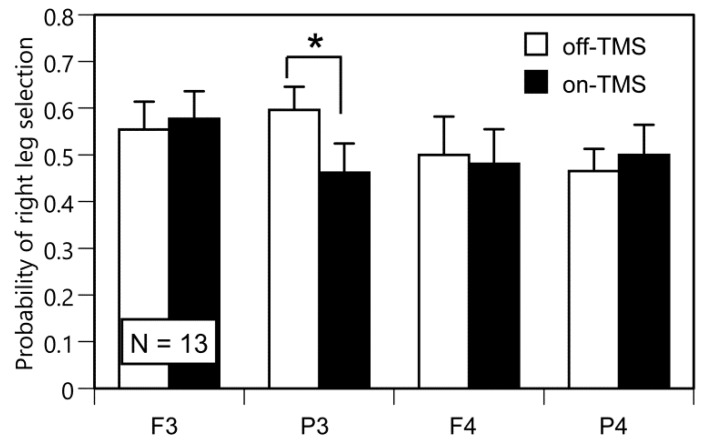
The probability of right leg selection for the tested gait initiation in the TMS session. The bars indicate the mean and the error bars indicate the standard errors of the mean. An asterisk indicates a significant difference between the on-TMS and off-TMS conditions (test of the simple main effect; *p* < 0.05). The probability of right leg selection was not significantly different from the theoretically equal probability of left and right leg selection (0.5) across the TMS sites in the off-TMS condition (one-sample *t*-test, *p* > 0.05).

**Figure 3 brainsci-10-00317-f003:**
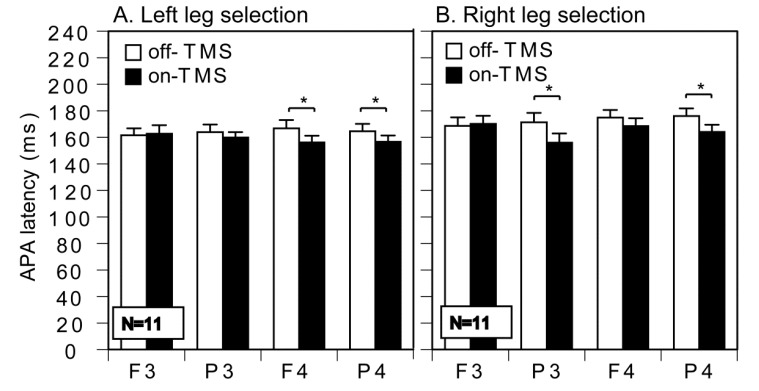
The APA latency of the tested gait initiation with the left (**A**) and right legs (**B**). The bars indicate the mean and the error bars indicate the standard errors of the mean. Asterisks indicate that the APA latency in the on-TMS condition was significantly shorter than that in the off-TMS condition (test of the simple main effect; *p* < 0.05).

**Figure 4 brainsci-10-00317-f004:**
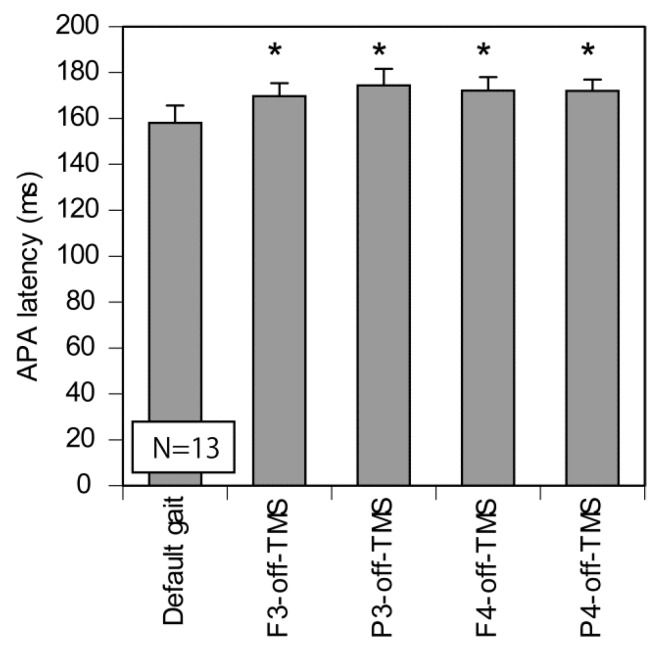
The APA latency of the tested gait initiation in the off-TMS condition and that of the default gait initiation. In the default gait initiation, the participants initiated the gait from an even feet position, but in the off-TMS conditions, they initiated the gait from the tested feet position in which the probability of the initial swing leg was about equal between the left and right. The bars indicate the mean and the error bars indicate the standard errors of the mean. Asterisks indicate that the APA latency in the tested gait initiation was significantly longer than that in the default gait initiation (multiple comparison test; *p* < 0.05).

**Figure 5 brainsci-10-00317-f005:**
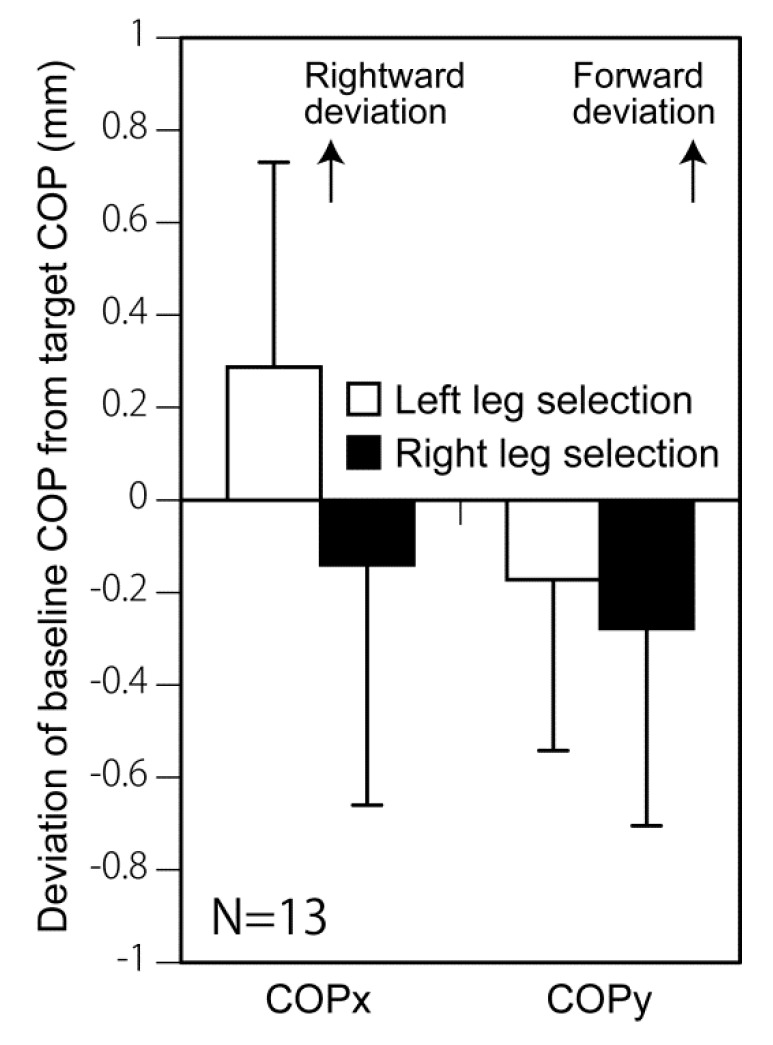
The deviation of the baseline COP from the target COP during gait initiation with the left and right legs in the off-TMS condition in the TMS session. The bars indicate the mean and the error bars indicate the standard errors of the mean. There was no significant difference in the baseline COP between gait initiation with the left leg and that with the right leg.

**Table 1 brainsci-10-00317-t001:** Measurements, number of participants included, and experimental session.

Measurement	N	Experimental Session		
		Default	Position	TMS
Preferred initial swing leg of default gait initiation	21	✓		
Correlation between WFQ and PRLS	15	✓		
Tested feet distance	15		✓	
Correlation between tested feet distance and PRLS	15	✓	✓	
TMS effect on the initial swing side	13			✓
Correlation between TMS effect on P3 and WFQ, PRLS, or tested feet distance	13	✓		✓
Equal probability of left and right leg swing	13			✓
Deviation from baseline COP (left vs. right leg swing)	13			✓
APA latency (left and right leg swing)	11			✓
APA latency (default vs. off-TMS)	13	✓		✓

Default: default gait initiation session; Position: tested feet position session; TMS: TMS session; WFQ: Waterloo Footedness Questionnaire; PRLS: probability of right leg selection; APA: anticipatory postural adjustment.

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
