# Peer review of "The Left Posterior Parietal Cortex Contributes to the Selection Process for the Initial Swing Leg in Gait Initiation"

_brainsci, 2020, doi:10.3390/brainsci10050317_

Round 1

Reviewer 1 Report

In the present study Hiraoka et al. explore the idea that activity in the left parietal cortex (PPC) contributes to side selection during gate initiation. They use transcranial magnetic stimulation (TMS) to focally disrupt the cortex at four different regions. The comparisons are made between conditions with the TMS stimulus on or off and between the effect of the four locations tested. The manuscript documents side preference during gait initiation, the center of pressure (COP) distribution right before gait initiation and the latency of anticipatory postural adjustment (APA). Given the known function of the PPC in attentional networks, executive function, and decision making, I approached the manuscript with great interest, and I think the question the authors ask has both scientific merit and is interesting to the community. Unfortunately, I personally found the manuscript very difficult to read, in part due to often unclear structure and the frequent uninterpretable sentences (e.g.: Line 234), and in part due to some inconsistencies in the use of phrases (e.g.: “swing leg side” vs. “swing side leg” vs “initial swing side” vs “side of legs”). While I understand that this is a subjective matter, the manuscript could benefit from some further editing. More importantly, I have doubts about the statistical approach and sample size representing the central finding of this manuscripts. I am fully aware that during these difficult times increasing the sample size in a study with human subjects is problematic but in its current form the data does not support the authors claims. I will detail the issues below.

Major comments:

  • In Figure 2, the authors compare the probability of initiating gate with the right leg under off- and on-TMS conditions at 4 sites. These conditions are obviously linked (performed in the same person in the same experimental blocks) and the ANOVA tests show that TMS has no effect and that none of the measurements are different from chance level (0.5). Yet, the authors then compare off- and on-TMs conditions separately using t-test. This seems statistically inappropriate, multiple comparisons post-hoc tests on individual pairs following an ANOVA test should be used. In fact, the probability of right leg selection in P3 off-TMS condition is clearly higher than the average probability of right leg selection throughout the whole experiment. This might explain why the on-TMS chance is significantly different (in a t-test) from off-TMS P3, while the average on-TMS P3 does not appear to be different from the chances in other testing sections. Essentially, Figure 2 suggests that the finding is simply due to small sample size and large variance between subjects and testing blocks.
  • The idea of the study being under-powered due to small sample size is supported by the fact that that the COP data in Figure 3 shows very high variance and yet the statistical comparison in the COPx direction is approaching significance at p=0.091, suggesting that a few extra subjects may reveal significant differences. Given that the results critically depend on the COP not predetermining the initial swing side, confidence in this data set is of high importance.
  • Throughout the manuscript the methodology is predominantly based on manual handling by the experimenters. This on its own right is not an issue, I am perfectly confident in the ability of the authors. However, I am missing critical experimental design elements for the reduction of unconscious biases and a thorough documentation of the experimental details. This includes:
    1. Were any attempts made to control for experimenter error in holding the TMS over the target region? Webcam footage or snapshots for example? Simply referencing that manual targeting has worked for other laboratories, thus the targeting is accurate in this case as well is hardly satisfactory.
    2. Is there a record of the COP related to baseline at the moment of trial initiation? Currently we are told in the Methods that the COP is within 1cm of the previously measured target, this accuracy somehow increases to 1mm in the Discussion, but there is no evidence of this being measured anywhere.
    3. Were the experimenters blinded to the experimental conditions? Given the amount of manual technique and experimenter involvement in the study, experimenter bias could not be avoided without conducting the research blind. This is not described in the manuscript.

Minor comments:

  • Please define all abbreviations during their first use (including the Abstract)
  • Please better define locations P3, P4, F3 and F4, specifically F3 and F4 locations are not described anywhere in the manuscript.
  • Was the marked swim cap re-used between participants or were the TMS target locations determined for each subject? How precise is such targeting?
  • It is unclear if the baseline COP was established from a single trial (Line 104) or once for each of the 10 repeats. If only once per experiment, it would be important to know what the variance of this is within a subject and how this variance could affect the experimental outcome.
  • Trial structure: were the locations and TMS on/off trials randomized or delivered in blocks?
  • Although the methods tell us that the subjects were told not to mind the experimental conditions, Figure 5 indicates that in fact, the experimental conditions had significant effects on the participants, including influencing the measure parameters. This could be a major confound of the entire study.
  • Although the statistical analysis shows “significance” in Figure 4, comparison to the literature to see if a 5-6% change in APA latency is meaningful at all would benefit the discussion?
  • The authors spend a lot of effort explaining that the coil click does not influence their results. This seems unnecessary considering that (1) the subjects wore headphones, (2) the simultaneous auditory cue was much louder than the TMS click, and (3) presumably the subjects were blind to the TMS on or off condition. However, TMS is known to induce discomfort in some subjects, thus the subjects could be influenced by TMS induced factors other than the auditory click, this does not seem to be discussed in the manuscript.

Author Response

Summary of revision

We greatly appreciate very important suggestions on our manuscript. We have carefully read the comments and revised our manuscript. In addition, a native English editor edited our revised manuscript. Followings are the summary of our revisions. The revised sentences were highlighted in the text. We would like to revise again if further revisions are needed.

Reviewer #1

(Comment)

Unfortunately, I personally found the manuscript very difficult to read, in part due to often unclear structure and the frequent uninterpretable sentences (e.g.: Line 234), and in part due to some inconsistencies in the use of phrases (e.g.: “swing leg side” vs. “swing side leg” vs “initial swing side” vs “side of legs”). While I understand that this is a subjective matter, the manuscript could benefit from some further editing.

(Response)

Thank you for the comment. We totally revised the manuscript, and a native English editor edited our manuscript. The phrase “initial swing leg side” was replaced with “initial swing side”. The phrase “preferred initial swing side” or “initial swing limb” was replaced with “preferred initial swing leg”. In some cases, the phrase “initial swing side” was replaced with “initial swing leg”.

(Comment)

In Figure 2, the authors compare the probability of initiating gate with the right leg under off- and on-TMS conditions at 4 sites. These conditions are obviously linked (performed in the same person in the same experimental blocks) and the ANOVA tests show that TMS has no effect and that none of the measurements are different from chance level (0.5). Yet, the authors then compare off- and on-TMs conditions separately using t-test. This seems statistically inappropriate, multiple comparisons post-hoc tests on individual pairs following an ANOVA test should be used.

(Response)

As the reviewer indicated, there were no significant main effects. In spite of that, there was a significant interaction between the two main effects (TMS site and TMS). Because of that, the simple main effect was tested for each level of another main effect. That is, the test of simple main effect was conducted to test the main effect of the TMS in each TMS site. A significant simple main effect of the TMS was found only in the P3. We did not conduct multiple t-test as the post-hoc test of the ANOVA. We reorganized and revised the statistics section for clarity (lines 217-239).

(Comment)

In fact, the probability of right leg selection in P3 off-TMS condition is clearly higher than the average probability of right leg selection throughout the whole experiment. This might explain why the on-TMS chance is significantly different (in a t-test) from off-TMS P3, while the average on-TMS P3 does not appear to be different from the chances in other testing sections. Essentially, Figure 2 suggests that the finding is simply due to small sample size and large variance between subjects and testing blocks.

(Response)

The figure seems to be as the reviewer indicated. However, the view indicated by the reviewer is not supported by the statistics; the test of the simple main effect following the two-way ANOVA found that the simple main effect of the TMS site in the off-TMS condition was insignificant. In spite of that, as the reviewer indicated, we can not rule out a possibility that small sample size may have affected our present finding. Thus, we discussed the small sample size in Limitation section in Discussion (lines 558-566).

(Comment)

The idea of the study being under-powered due to small sample size is supported by the fact that that the COP data in Figure 3 shows very high variance and yet the statistical comparison in the COPx direction is approaching significance at p=0.091, suggesting that a few extra subjects may reveal significant differences. Given that the results critically depend on the COP not predetermining the initial swing side, confidence in this data set is of high importance.

(Response)

The COP was strictly controlled through triggering the start cue only when the COP was within ± 1 cm of the baseline COP (lines 112-116, 134-136, 170-171). In addition, the average deviation of the COP from the baseline COP was within 1 mm (line 307), and the COP deviation in the trials with gait initiation with the left leg and that with gait initiation with the right leg was insignificant (lines 307-309). Thus, it is unlikely to speculate that the deviation of the COP is the cause of the TMS effect (lines 399-416).

(Comment)

Throughout the manuscript the methodology is predominantly based on manual handling by the experimenters. This on its own right is not an issue, I am perfectly confident in the ability of the authors. However, I am missing critical experimental design elements for the reduction of unconscious biases and a thorough documentation of the experimental details.

(Response)

We agree with the reviewer’s suggestion that unconscious bias is the important independent variable on the leg selection. To rule out this, we did not inform the patients that we were observing the initial swing leg (lines 108-109, 143-144, 167-168), and asked them not to mind the initial swing leg (lines 107-108, 166-167). In addition, we asked the participants not to mind the coil over the head (lines 166-167). Indeed, as shown in Results, the test of simple main effect showed no significant difference in the probability of the right leg selection among the TMS sites in the off-TMS condition (lines 287-289). This indicates that the coil position did not influence the leg selection.

(Comment)

This includes:

Were any attempts made to control for experimenter error in holding the TMS over the target region? Webcam footage or snapshots for example? Simply referencing that manual targeting has worked for other laboratories, thus the targeting is accurate in this case as well is hardly satisfactory.

(Response)

The coil was held by an experimenter manually, but any evidence proving that the coil was positioned precisely over the TMS site. This was an weakness of this study. We added this in Limitation section in Discussion (lines 557-558).

(Comment)

Is there a record of the COP related to baseline at the moment of trial initiation? Currently we are told in the Methods that the COP is within 1cm of the previously measured target, this accuracy somehow increases to 1mm in the Discussion, but there is no evidence of this being measured anywhere.

(Response)

Thank you for the comment. A trigger of the start cue was given when the COP was within ± 1 cm of the target COP (lines 112-116, 134-136, 170-171). As the results, mean baseline COP deviated from the target COP was 1 mm (line 307). This small deviation was the result of the cancellation of the positive and negative deviations across the trials. The deviation of the COP from the baseline in the time window 0-100 ms before the start cue is shown in Results (3.4.) and in Fig. 3.

(Comment)

Were the experimenters blinded to the experimental conditions? Given the amount of manual technique and experimenter involvement in the study, experimenter bias could not be avoided without conducting the research blind. This is not described in the manuscript.

(Response)

Thank you for the comment. In the TMS session, the coil position and TMS condition were randomly altered trial by trial (lines 194-195). The presence or absence of the TMS in the forthcoming trial was not informed to the experimenter who held a coil and to the participants (lines 191-192). Thus, the presence or absence of the TMS in the forthcoming trial was not predictable for the participants and for the experimenter who held the coil.

(Comment)

Please define all abbreviations during their first use (including the Abstract)

Thank you. The phrases were spelled out the first time the abbreviation is used (e.g., WFQ and APA) (lines 32, 82).

(Comment)

Please better define locations P3, P4, F3 and F4, specifically F3 and F4 locations are not described anywhere in the manuscript.

(Response)

Thank you for the comment. F3 and F4 correspond to the DLPFC (Herwig et al. 2003) or middle frontal gyrus (Homan et al. 1987). This was mentioned in Methods (lines 53-54).

(Comment)

Was the marked swim cap re-used between participants or were the TMS target locations determined for each subject? How precise is such targeting?

(Response)

Thank you. The TMS sites were marked over the swim cap by measuring the distance from the inion, nation, and ears in accordance with the international 10-20 system in each participant. We added this in Methods (lines 159-164).

(Comment)

It is unclear if the baseline COP was established from a single trial (Line 104) or once for each of the 10 repeats. If only once per experiment, it would be important to know what the variance of this is within a subject and how this variance could affect the experimental outcome.

(Response)

Thank you for the comment. As the reviewer indicated, we measured once to estimate the target COP before the experiment. This was a kind of calibration process to determine the default target COP position in the quiet stance. As the reviewer suggested, the COP in the standing must have small deviation. In spite of that, the target COP was same across the experiment, and thus, this must not have influenced the results.

(Comment)

Trial structure: were the locations and TMS on/off trials randomized or delivered in blocks?

(Response)

We did not take a block design. The TMS condition was randomly altered trial by trial in the session (lines 194-195). The presence or absence of the TMS in the forthcoming trial was not informed to the experimenter who held a coil and to the participants (lines 191-192). Thus, the presence or absence of the TMS in the forthcoming trial was not predictable for the participants and for the experimenter who held the coil.

(Comment)

Although the methods tell us that the subjects were told not to mind the experimental conditions, Figure 5 indicates that in fact, the experimental conditions had significant effects on the participants, including influencing the measure parameters. This could be a major confound of the entire study.

(Response)

Thank you for the comment. In the default gait initiation, they initiated gait from the even feet position (lines 102-105). In the other conditions, they initiated gait from the tested feet position, in which probability of the initial swing side was about equal between the left and right (lines 155-157). That is, for the trials in the left most bar, they initiated gait with the preferred swing leg, but in the other bars, the initial swing side was even between the legs. Through this analysis, we tested whether the competition between the motor process of the gait initiation with the left leg and that with the right leg influences the time taken to process the APA (lines 225-230, 333-339). The latency was longer for the off-TMS conditions comparing with the default gait initiation, supported our view (lines 339-343). This finding was discussed in 4.9 (lines 525-546). In order to help understanding, we revised the text and figure legend of the Fig. 5 (lines 350-352).

(Comment)

Although the statistical analysis shows “significance” in Figure 4, comparison to the literature to see if a 5-6% change in APA latency is meaningful at all would benefit the discussion?

(Response)

Change rate of the significant difference in the APA latency was 7%. In a previous study by Mizusawa and colleagues, the gait initiation with go trial, that with switch trial, and that with stop trial were tested, and the change rate was 8-10%. Thus, change in the central motor process may change the APA latency around 7-10%. This issue was additionally discussed in Discussion (lines 502-510).

(Comment)

The authors spend a lot of effort explaining that the coil click does not influence their results. This seems unnecessary considering that (1) the subjects wore headphones, (2) the simultaneous auditory cue was much louder than the TMS click, and (3) presumably the subjects were blind to the TMS on or off condition. However, TMS is known to induce discomfort in some subjects, thus the subjects could be influenced by TMS induced factors other than the auditory click, this does not seem to be discussed in the manuscript.

(Response)

Thank you for the important comment. Discomfort induced by TMS is unlikely because the TMS intensity was as low as 40% of maximal stimulator output (lines 93-94). Coil position is a possible factor associated with the finding. In spite of that, the test of the simple main effect showed that there was no significant effect of the coil position (TMS site) in the off-TMS condition (lines 287-289). Thus, the present finding did not support a view that coil position influenced the finding. This was discussed in 4.1 (lines 363-370).

Reviewer 2 Report

Please see the attachment for my comments.

Author Response

Summary of revision

We greatly appreciate very important suggestions on our manuscript. We have carefully read the comments and revised our manuscript. In addition, a native English editor edited our revised manuscript. Followings are the summary of our revisions. The revised sentences were highlighted in the text. We would like to revise again if further revisions are needed.

Reviewer #2

(Comment)

The introduction could use some work. Need to address the importance of the research question. The research aim and hypothesis are not clear presented.

(Response)

Thank you for the comment. We added the hypothesis of the present study at the end of the introduction (lines 75-77).

(Comment)

Methods

There is a lot of redundancy in the current methods, especially for the 2.3 and 2.4.

(Response)

Thank you for the comment. In 2.3., the paragraph was divided into three, and some revisions were made for clarity. In the paragraph 1 of 2.4, descriptions already mentioned in 2.3 were deleted to reduce redundancy.

(Comment)

These protocols are difficult to read to understand the flow. My suggestion would be adding an illustration for your research protocol, e.g., a flow chart would be a good candidate for Fig 1.

(Response)

Thank you for proposing a good idea. In our opinion, adding a table stating the measurements is better than flow chart. We added Table 1 and cited this table in Results (lines 249, 259, 273, 335).

(Comment)

For the cueing protocol, it appears that timing of acoustic cue delivery was random since it relied upon the baseline center of pressure met the target (within 1cm). This way, the participants could have missing the cue, as stated in the manuscript, some trials were discarded when the APA latency was over 300ms. This could be prevented by using a different cueing paradigm, for instance an imperative-go paradigm with a fixed timing interval between a warning and go cue. In addition, this allows us to ensure the participants are “prepared” to execute the task to a better extent.

(Response)

Thank you for the thought-provoking comment. There were very few trials in which the APA latency exceeded 300 ms. Thus, the deficit of attention unlikely affected the present findings. Warning cue with fixed interval between the warning and start cue may cause time preparation. In addition, if the experimenter triggers the warning cue instead of the start cue, the COP may out of the target range in the interval between the warning and start cues. We did not give warning cue to rule out those problems. A sentence explaining those was added in Methods (lines 118-120).

(Comment)

Regarding TMS, detailed stimulation parameters for TMS is missing in the methods.

(Response)

Thank you for the comment. A figure of eight coil was used (lines 91-92). Maximum intensity of the coil was 0.96 T (lines 92-93). The TMS intensity was 40% of the maximum stimulator output (lines 93-94). The coil was along with the rostro-caudal direction (lines 164-165). The direction of the electrical current in the brain was added (lines 164-165).

(Comment)

In addition, were both participants and experimenters blinded to the TMS-on and TMS-off?

(Response)

Yes, the experimenter holding the coil and the participants were not informed whether the TMS was given or not in the forthcoming trial (lines 191-192), and the on/off of the TMS was randomly changed trial by trial in a session (lines 194-195).

(Comment)

Was the TMS-off provided as a sham stimulation such that the participant would not be able to perceive the difference.

(Response)

In the off-TMS condition, TMS was not given. However, we confirmed that they could not perceive the coil click because the loud auditory start cue was given with the TMS (lines 153-155, 275-277). Thus, the coil click did not influence the finding.

(Comment)

For data analysis, since the aim/hypothesis is not explicitly provided, it’s hard to follow what tests should be performed here. First, it’s not clear what the dependent variables are for the Two-way repeated measures ANOVA.

(Response)

Thank you. We described the hypothesis of the one-way and two-way ANOVAs in Data analysis section (lines 218-222, 224-230).

(Comment)

Second, please state the rationale of performing One-way repeated measures ANOVA between TMS-off and default gait initiation.

(Response)

Thank you. We described the hypothesis of the one-way and two-way ANOVAs in Data analysis section (lines 218-222, 224-230).

(Comment)

In addition, some statistical tests were provided, but without useful information, for example, the paired t-test in lines 202-203 and the Pearson’s correlation in line 207 needs more details.

(Response)

Thank you. For the correlation coefficient, the interpretation of the results was mentioned in Results (lines 252-253, 261-263, 293-294). Detailed explanation of the t-test was mentioned in Methods (lines 232-233). In addition, in Results, we added “paired t-test” in the place where the p-value for the t-test appeared (line 309).

(Comment)

Results

3.1, correlation stats were presented; however, I believe neither the probability and WFQ scores are normally distributed and therefore a spearman’s should be used here.

(Response)

Thank you for the comment. In the revised manuscript, we conducted Spearman’s rank correlation coefficient instead of Pearson’s correlation coefficient (lines 237-238).

(Comment)

A scatter plot with probability and WFQ scores should be provided along with the stats to better visualize the relationship. to The authors performed multiple mixed models to draw the conclusion that gait stability was affected differently with different cueing approach. Similar for 3.2, a scatter plot with tested feet distance and probability should be provided.

(Response)

Thank you for the comment. We provided the scatter plot for the correlation coefficient with statistical significance; i.e., correlation coefficient between the PRIS and tested feet distance (Fig. 1D). In other correlation coefficients, data plots were not provided because those were statistically insignificant.

(Comment)

3.3, Six out of 21 participants (about 28%) were excluded in the analysis. This should be further discussed and presented as a limitation.

(Response)

Thank you for the comment. We discussed this issue in 4.10 (lines 548-554).

(Comment)

Again, it’s convincing to treat the probability of leg selection as normally distribution. In addition, the non-significant results could be due to an insufficient power. Please provide a power analysis or effect size estimation.

(Response)

Thank you for the comment. We added effect size in the results of the ANOVA.

(Comment)

Discussion

The order of the paragraph seems misplaced and the flow seems to be disconnected. The most likely explanation should be discussed after the 1st paragraph of the discussion.

(Response)

Thank you for the comment. The order of the paragraph in Discussion was made to rule out the possible causes affecting the leg selection first, and then most likely mechanism was discussed secondly. We believe that this order is clearer than the order in which most likely mechanism is discussed first.

(Comment)

The section 4.1 to 4.4 read more like a potential limitation/explanation, but already addressed by the authors.

(Response)

As the reviewer indicated, some descriptions in the section 4.1 and 4.4 have already been mentioned in Introduction. Nevertheless, in Discussion, we discussed those issues with the present findings to confirm that our major concerns were ruled out or not.

(Comment)

There are a few studies using startle paradigm to probe the movement initiation. This line of research should be discussed, but it’s completely missing in the current manuscript, such as MacKinnon et al., 2007; Nonnekes et al., 2014.

(Response)

Thank you for introducing important articles. We cited an article by MacKinnon and colleagues in the text (lines 178-181). We found articles by Nonnekes published in 2014, but we could not find the articles related to the present study (all of the articles in 2014 were related to the freezing of gait in patients with Parkinson's disease).

(Comment)

In addition, it would be beneficial to see results during not only postural preparation but also execution of the 1st step (e.g., step time & step length).

(Response)

Thank you for the comment. We did not measure the time of the heel contact and heel off in the present study. We are sure that our hypotheses are supported by the present data without the step time or step length.

(Comment)

Grammar: line 54: “must been”

(Response)

Thank you for the comment. This phrase disappeared through the revision process.

Round 2

Reviewer 1 Report

The authors addressed most main concerns in the revised manuscript which is a marked improvement over the original version. Language editions are much welcome, it has noticeably enhanced the readability of the text. I also appreciate the clarifications in the statistics section. While my main concern of limited sample size remains, the authors did an admirable job at pointing out this and other potential weaknesses of their study.

Author Response

Summary of revision

We greatly appreciate very important suggestions on our manuscript. We have carefully read the comments and revised our manuscript. Followings are the summary of our revisions. The revised sentences were highlighted in the text. We would like to revise again if further revisions are needed.

Reviewer #1

The authors addressed most main concerns in the revised manuscript which is a marked improvement over the original version. Language editions are much welcome, it has noticeably enhanced the readability of the text. I also appreciate the clarifications in the statistics section. While my main concern of limited sample size remains, the authors did an admirable job at pointing out this and other potential weaknesses of their study.

(Response)

Thank you for the thorough review of our manuscript. Your comments on our manuscript greatly helped our revision process.

Author Response

Summary of revision

We greatly appreciate very important suggestions on our manuscript. We have carefully read the comments and revised our manuscript. Followings are the summary of our revisions. The revised sentences were highlighted in the text. We would like to revise again if further revisions are needed.

Reviewer #2

(Comment)

Results

Section 3.1 is not related to the introduction or the main research question at all. In addition, section 3.2 and 3.4 don’t fit in the main results very well. 3.2 is an important methodological concern needed to be addressed (as stated in the intro), but this is not related to the primary research question. This can be reported in the methods (3.2 ± 1.4 cm) or even in supplementary materials. Same as the section 3.4, it’s not set up in a way that this result is directly linked to the research question. Both 3.2 and 3.4 read as if these are results for arguments against alternative explanations for current findings. Please remove some sections or reorganize the results (e.g., at least move 3.2 & 3.4 to the end) and keep the sections directly linked to the question, such as 3.3 and 3.5.

(Response)

Thank you. We agree with the reviewer’s suggestion. We moved 3.2 and 3.4 to the end of the Results. At the same time, the fig. 3 was moved to fig. 5, and figs 3 and 4 were moved to figs 2 and 3 respectively.

(Comment)

The range for tested feet distance is quite large (0 to 17 cm), which implicated the base of support is quite different among the subjects. This could be a potential confounding factor since the findings could be driven by the potential outlier with a large base of support. Thus, I would recommend the tested foot distance could be treated as a covariate in the repeated measures ANOVA.

(Response)

Thank you for the important comment. To address the reviewer’s concern, we made another additional analysis. Our additional data analysis showed that the tested feet distance was not significantly correlated with the TMS effect over the P3 (lines 285 - 289). Thus, the reviewer’s concern is unlikely. If we add the tested feet distance as another main effect, ANCOVA or three-way ANOVA for the small sample size must be conducted, causing lesser sensitivity of the statistics and increasing the risk of the type II error. Thus, we did not include the tested feet distance as the main effect.

(Comment)

Discussion

I still disagree with the flow in discussion. Again. Please discuss the “data” first. There is no results/analysis directly related to section 4.1, 4.2. This is still very hard to read. The audience would like to see discussion with the results directly linked to research questions, i.e., section 3. 3 TMS affect leg selection during gait initiation, and section 3.5 TMS effect on APA latency

(Response)

Thank you for the comment. We agree with the reviewer’s suggestion. We reorganized Discussion. We firstly discussed the cortical area (left PPC) contributing to the selection process of the initial swing leg (4.1). Secondly, before discussing the APA latency, the neural process in the reaction time was mentioned (4.2). Then, we discussed most likely explanation of the effect of TMS over the P3 based on the finding on the APA latency (4.3.). After this section, unlikely explanations (coil position, coil click, weight distribution, and unequal probability of the initial swing leg, and preferred initial swing leg) were discussed (4.4 – 4.8).

(Comment)

The sentences are completely the same in lines 26 and 57 -58. Please rephrase either one.

(Response)

Thank you for the comment. We rephrased the secondly appearing sentence (lines 57-58).

(Comment)

Figure 1D bottom left two data points should be gray-filled dots.

(Respopnse)

Thank you for the comment. The data points were painted in gray.

(Comment)

Line 241: change wording “function” to “relationship”.

(Response)

Thank you. We phrased “as relation with” instead of “as function of” (line 241).
